# Paravertebral Muscle Training in Patients with Unstable Spinal Metastases Receiving Palliative Radiotherapy: An Exploratory Randomized Feasibility Trial

**DOI:** 10.3390/cancers11111771

**Published:** 2019-11-11

**Authors:** Tanja Sprave, Friederike Rosenberger, Vivek Verma, Robert Förster, Thomas Bruckner, Ingmar Schlampp, Tilman Bostel, Thomas Welzel, Sati Akbaba, Tilman Rackwitz, Nils Henrik Nicolay, Anca-Ligia Grosu, Joachim Wiskemann, Jürgen Debus, Harald Rief

**Affiliations:** 1Department of Radiation Oncology, University Hospital of Freiburg, Robert-Koch-Strasse 3, 79106 Freiburg, Germany; nils.nicolay@uniklinik-freiburg.de (N.H.N.); anca.grosu@uniklinik-freiburg.de (A.-L.G.); 2Department of Medical Oncology, National Center for Tumor Diseases (NCT), Heidelberg University Hospital, 69120 Heidelberg, Germany; friederike.rosenberger@nct-heidelberg.de (F.R.); joachim.wiskemann@nct-heidelberg.de (J.W.); 3Department of Radiation Oncology, Allegheny General Hospital, Pittsburgh, PA 15212, USA; vivek333@gmail.com; 4Department of Radiation Oncology, University Hospital Zurich, Raemistrasse 100, 8091 Zurich, Switzerland; robert.foerster@usz.ch; 5Department of Medical Biometry, University Hospital of Heidelberg, Im Neuenheimer Feld 305, 69120 Heidelberg, Germany; bruckner@imbi.uni-heidelberg.de; 6Department of Radiation Oncology, Heidelberg Institute of Radiation Oncology (HIRO), Im Neuenheimer Feld 280, 69120 Heidelberg, Germany; ingmar.schlampp@med.uni-heidelberg.de (I.S.); thomas.welzel@med.uni-heidelberg.de (T.W.); sati.akbaba@med.uni-heidelberg.de (S.A.); tilman.rackwitz@med.uni-heidelberg.de (T.R.); juergen.debus@med.uni-heidelberg.de (J.D.); 7Department of Radiation Oncology, University Medical Center of the Johannes Gutenberg-University Mainz, 55131 Mainz, Germany; tilman.bostel@unimedizin-mainz.de; 8Heidelberg Institute of Radiation Oncology (HIRO), German Cancer Research Center (DKFZ), 69120 Heidelberg, Germany; 9Radiooncological Practice Bad Godesberg, Waldstrasse 73, 53177 Bonn Bad-Godesberg, Germany; harald.rief@gmx.at

**Keywords:** spine metastases, unstable, training, palliative radiotherapy

## Abstract

*Background*: Isometric paravertebral muscle training (IPMT) may improve mobility, pain, and quality of life (QOL) in cancer patients with spinal metastases. However, this regimen remains unproven in patients with unstable spinal metastases (USM), a population at high risk for clinical exacerbation with such interventions. Thus, we conducted this exploratory, non-blinded, randomized controlled trial (NCT02847754) to evaluate the safety/feasibility of IPMT and secondarily assess pain, bone density, pathologic fracture rate, and QOL. *Methods*: All patients had histologically/radiologically confirmed USM (per Taneichi score) and underwent non-operative management with 5–10 fractions of palliative radiotherapy (RT). Randomization (1:1) groups were IPMT (intervention, INT) or muscle relaxation (control, CON); both lasted 15 min/day and started concurrently with radiotherapy. The primary endpoint was feasibility (completion of training programs three months post-RT). Secondary endpoints were pain response (Visual Analog Scale) and opioid consumption, bone density and pathologic fracture rate, and QOL (European Organization for Research and Treatment of Cancer, EORTC questionnaires). *Results*: Sixty patients were randomized and 56 received protocol therapy. Mean survival in both groups was 4.4 months. There were no adverse events with either training regimen. Altogether, ≥80% of the planned sessions were completed by 55% (n = 16/29) in CON and 67% (n = 18/27) in INT. Regarding the post-radiotherapy home-based training, ≥80% of planned sessions were completed by 64% (n = 9/14) of the INT cohort. There were no differences in pain scores, opioid consumption, or bone density between arms (*p* > 0.05 for all). No difference was observed between groups regarding new pathological fractures (INT: n = 1 vs. CON: n = 3) after three months (*p* = 0.419). There were no QOL differences between arms (all parameters *p* > 0.05). *Conclusions*: IPMT is potentially feasible for high-risk USM patients. Future trials adequately powered for relevant endpoints are thus recommended.

## 1. Introduction

The spine is a very common area of metastatic disease [1,2]; thus, activities of daily living and quality of life (QOL) can be markedly hampered in these patients. Spinal metastases can be categorized as stable or unstable, based on several factors such as tumor burden and location, symptomatology, and several parameters on imaging [3].

Whereas stable spinal metastases are often treated with palliative radiation therapy (RT) alone, management of unstable spinal metastases (USM) represents an interdisciplinary challenge [4,5,6,7]. Although surgical therapy is commonly performed, many patients with metastatic cancer are not surgical candidates for several reasons. Thus, palliative RT remains an effective treatment option for spine instability and pain [8,9,10,11,12,13]. Conservative treatment often involves patient immobilization, most commonly by utilizing an orthopedic corset or with prolonged bedrest.

In order to improve QOL in cancer patients, numerous short- and long-term effects of targeted physical training measures have been reported, with practical and clinically meaningful improvements in pain and mobility [14,15,16,17,18]. Specifically, additional isometric paravertebral muscle training (IPMT) may allow for strengthening paraspinal muscles and improving mobility, pain, and QOL. In a previous randomized trial for stable spine metastases, IPMT (concomitant with palliative RT) affected some of the aforementioned endpoints and did not increase the pathologic fracture rate [19,20,21,22,23].

Despite these encouraging results, this regimen remains unproven for USM; as a result, most prospective trials in this population remain reluctant to implement such interventions, since IPMT in this high-risk population could lead to clinical exacerbation, including increasing the risk of pathologic fracture.

To address this knowledge gap, we conducted this exploratory randomized study, the first of its kind to date, to evaluate the feasibility of IPMT (as compared to muscle relaxation) and secondarily determine effects on pain, bone density, pathologic fracture rate, and quality of life [24].

## 2. Materials and Methods

### 2.1. Design and Patient Population

This exploratory randomized controlled trial (NCT02847754) was approved by the Heidelberg University Independent Ethics Committee (S-223/2016) (Appendix A). The goal of this study was to evaluate the feasibility of paravertebral muscle-training exercises (interventional group, INT), as compared to muscle relaxation (control group, CON), in patients with USM receiving palliative RT [24]. The randomization procedure was carried out by a central office; a block randomization approach (block size of 6) was utilized.

From December 2016 to November 2018, 60 patients with histologically confirmed cancer and USM of the thoracolumbar segments were considered for this study. USM was defined based on computed tomography (CT) and/or magnetic resonance imaging (MRI) assessment based on the well-recognized Taneichi score [19,22,25]. Surgical intervention to the area of USM was not allowed, mainly because the degree and extent of surgical intervention (based on tumor location) would add a major confounding factor regarding the ability to postoperatively perform paravertebral muscle training exercises in a uniform time frame. As such, this study included inoperable cases (secondary to disease extent, or medical contraindications) as well as subjects who refused surgery.

Other inclusion criteria were ages 18–80 years, Karnofsky performance score ≥ 70, ability to provide written informed consent, and an established indication for palliative RT. In order to address potential confounding by bisphosphonates or anti-RANKL agents, one of these compounds was required to be delivered if the patient was not already receiving one such agent. Exclusion criteria were previous RT or surgery to the given irradiation site, spinal cord compression according to the Bilsky score, myeloma/lymphoma histology, involvement of the cervical spine, and/or inability/refusal to complete the given exercise regimen.

### 2.2. Interventions

Complete details of IPMT (INT group) are presented elsewhere [24]. Briefly, these consisted of exercises (1:1 supervised by exercise physiologists or physical therapists) performed once daily, starting on the first day of palliative RT and continuing for the entire RT period. Following RT completion, subjects continued the same exercises three times per week (corroborated by a daily log) in a home-based manner for another three months. The overall exercise regimen was estimated to take 15 min per day and consisted of isometric exercises in four positions: “all fours” (each extremity stretched separately), “plank”, “swimming” (toes kept on the floor), and upright with an elastic band tightened in front of the trunk. The holding time for each position was 20 s initially, and increased from session to session when feasible. The exercises were performed without a corset.

Muscle relaxation (CON group) was also performed for an estimated 15 min (once daily) during palliative RT as above. These exercises comprised of progressive muscle relaxation for the face, arms, abdomen, and legs. The back was excluded to avoid training effects on the paravertebral muscles. Muscle relaxation was similarly performed with 1:1 supervision and could voluntarily be continued following completion of RT (corroborated by an audio CD).

Palliative RT was delivered in either three-dimensional conformal (3DCRT) or intensity-modulated (IMRT) techniques. Stereotactic body radiation therapy (SBRT) was not allowed for this study. For both techniques, the involved vertebra was treated to a dose of 20 Gy in 5 fractions or 30 Gy in 10 fractions. If IMRT was utilized, simultaneous integrated boosting (SIB) was allowed to 30 Gy for a 5-fraction regimen and 40 Gy for a 10-fraction regimen. Treatment planning was based on parameters in the Radiation Therapy Oncology Group (RTOG) 0631 study [26] and QUANTEC [27] recommendations. Position verification was carried out weekly before radiotherapy by kilovoltage cone-beam CT and before each fraction by orthogonal portal images being compared with digitally reconstructed radiographs from the planning CT.

### 2.3. Endpoints

Both the primary and secondary endpoint-related parameters were measured at the start of RT (t_0_), the end of RT (t_1_), three months post-RT (t_2_), and 6 months post-RT (t_3_). During therapy, the treating clinicians documented these parameters, but diaries were used to document patient-reported information subsequently.

Because performing IPMT for USM risks clinical exacerbation (including increasing the risk of pathologic fracture) the primary endpoint of this randomized investigation was feasibility, which referred to completion of the training program at three months following the end of RT. The total number of completed and aborted/canceled training units and adverse events during training was recorded.

The initial secondary endpoint was the pain score, as measured by the Visual Analog Scale (VAS). Pain level was measured by subjective patient reporting on the VAS scale with a range of 0–100. During clinical examination by the study physician, neuropathic pain was also monitored, as well as pain medication usage (opioid usage was converted into an oral morphine equivalent dose (OMED); non-opioid analgesics were also recorded).

Additional secondary endpoints were bone density and pathologic fracture rate. Bone density was assessed in the irradiated (and unirradiated) vertebral bodies by a single physician with CT imaging (Siemens Somatom Sensation Open, Siemens, Erlangen, Germany) and Syngo Osteo CT workstation in manually selected regions of interest; Hounsfield units were used for bone density measurements. Pathologic fractures were diagnosed by means of CT and/or MRI imaging and comparing to baseline imaging tests. New fractures were, by definition, not present on initial imaging, whereas progressive fractures referred to visibly increasing size and/or number of fracture gaps, dislocation of fracture fragments, or increasing sintering of the compression fracture (if applicable).

The final secondary endpoint was QOL, assessed using the EORTC QLQ BM22 questionnaire, specially designed for patients with bone metastases. This module (range 0–100) comprises of 22 items and four scales for the measurement of pain in various parts of the body (painful sites), pain characteristics (persistent pain, recurrent pain), functional impairment (occurrence of pain when performing different activities, interference with everyday activities), and psychosocial aspects (family, worries, hope) [28]. Fatigue was assessed using the EORTC QLQ FA13 (range 0–100) module, encompassing 13 items and five scales for measuring cancer-related fatigue [29], with subscales covering physical fatigue, emotional fatigue, cognitive fatigue, interference with daily life, and social sequelae. Emotional distress was assessed using the QSC-R10 (range 0–50) questionnaire, which is a reliable questionnaire for determining emotional distress and anxiety in cancer patients [30].

### 2.4. Statistical Analysis

Owing to the exploratory nature of this trial and lack of literature-based reference values, a complete power calculation was not possible; however, with 30 patients in each group, it was possible to detect a standardized mean-value effect of 0.8 with 80% power at a significance level of 0.05 [24].

All statistical analyses were done using SAS software Version 9.4 or higher (SAS Institute, Cary, NC, USA). All variables were analyzed descriptively by tabulation of the measures of the empirical distributions. According to the scale level of the variables, means and standard deviations (SD) or absolute/relative frequencies, respectively, were reported. Additionally, for variables with longitudinal measurements, the time courses of individual patients were summarized by treatment groups. Descriptive *p*-values of the corresponding statistical tests comparing the treatment groups were reported. The VAS was adjusted for concurrent medications. Analysis of covariance (ANOVA) with repeated measurements, with treatment group as a factor, and pain medication as a covariate, were done. The Wilcoxon signed-rank test was used to detect possible differences between groups after 3 and 6 months. Graphical visualization includes the mean course over time. Finally, we compared the groups for overall survival, using Kaplan–Meier estimates and log-rank tests. Overall survival (OS) was defined as time from randomization until death, or censored at last contact.

## 3. Results

### 3.1. Patient Details

Sixty patients were randomized, and 56 patients started on protocol-based management (Figure 1). One patient (CON) was removed for rapid clinical deterioration from cancer progression, one (INT) for new-onset jugular vein thrombosis, one (INT) for withdrawal of consent, and the final (INT) for severe motion-dependent therapy-resistant pain symptoms.

Baseline characteristics were balanced between the two arms (Table 1). Most patients had thoracic spine disease, and statistical similarities were noted regarding the location of distant metastases, oncologic therapy, and pain medication utilization (*p* > 0.20 for all). Of note, ten patients in the INT cohort and 14 subjects in the CON group initially wore an orthopedic corset (*p* = 0.396). Additionally of note, the Spinal Neoplastic Instability Score (SINS) [3] in INT was significantly higher as compared to CON (12.0 vs. 10.3, *p* = 0.007), whereas the Mizumoto score was similar (5.0 vs. 5.5, *p* = 0.260).

The median follow-up was not yet achieved in both arms. Eleven patients (40.7%) in the INT group died of disease within three months of RT, versus six patients (20.7%) in the CON group. The mean OS was 4.4 months for both groups (*p* = 0.839) (Figure 2).

### 3.2. Tolerance of Therapy/Feasibility

RT was altogether tolerated well. No patient in either arm experienced grade ≥3 acute or late events according to the Common Terminology Criteria for Adverse Events v.4.03.

During the supervised training (t_0_–t_1_) there were no adverse events with protocol therapy. In the CON arm, 16 patients (55%) completed ≥80% of the planned relaxation sessions; the remainder were unable owing to deterioration in the general condition or clinical (non-protocol-related) complications. In INT, 18 (67%) patients completed ≥80% of the planned training sessions. The mean total number of completed training units was 7.8 (SD 3.3), and the mean number of potentially feasible units was 10.1 (SD 2.1).

Similarly, no adverse side effects were reported during post-radiotherapy home-based training (t_1_–t_2_). The specified number of home training sessions was 36 (3× weekly over 12 weeks). In the INT arm, 14 participants were lost to follow-up during the period of t_1_–t_2_; of these subjects, 11 died and 3 were unknown. From t_1_ to t_2_, ≥80% of planned sessions were completed by 64% (9/14) of patients. In INT, 14 analyzed participants completed 39.6 (SD 21.1) of the prescribed 36 training sessions.

### 3.3. Pain Response

No difference in pain response was observed between the two groups after 3 and 6 months (Table 2).

There were also no differences in OMED consumption at the end of RT (t_1_) (*p* = 0.958) and three months (t_2_) following RT (*p* = 0.666). There were no statistical differences in neuropathic pain between both arms at 3 (*p* = 0.826) or 6 months (*p* = 0.965).

The covariance analysis of the OMED consumption in the period t_0_–t_2_ showed no significant influence on pain level (*p* = 0.120). The covariate evaluation of the interaction between group and time showed no significance, because the temporal changes were parallel (*p* = 0.970). Also, the group effect was not significant (*p* = 0.316). The pain response in the period t_0_–t_2_ showed a clear temporal dependence (*p* = 0.009) (Figure 3).

### 3.4. Bone Density and Pathologic Fractures

There were no differences in bone density between arms at 3 (*p* = 0.826) or 6 months following RT completion (*p* = 0.965). Within the CON group, from t_0_ to t_2_ there was a significant increase in bone density (*p* = 0.006) (Table 3).

At initial presentation, there was a trend towards more pathologic fractures in the INT arm (n = 17, 63% vs. CON: n = 11, 39%, *p* = 0.079). No pathologic fractures in either arm were de novo; 1/14 and 3/18 cases were progressive in INT and CON, respectively (*p* = 0.419). There were no differences at 6 months (*p* = 0.243). Of note, no cases of salvage surgery for pathologic fractures were necessary in either arm.

Additionally, there did not seem to be differences in 3-month pathologic fractures based on use of an orthopedic corset (31% vs. 35%, *p* = 0.673).

### 3.5. Quality of Life

In the INT group, the QOL parameter specifically for contemplation of painful sites had improved from initial presentation to the end of RT (*p* = 0.050), with a further positive trend between 3 and 6 months (*p* = 0.057). However, there was no evidence of treatment effect between t_0_–t_2_ (*p* = 0.478) or t_0_–t_3_ (*p* = 0.753) (Table 4, Table 5 and Table 6).

At all recorded time points, there were no significant QOL differences between groups, including pain characteristics, functional impairment, or psychosocial aspects (*p* > 0.05 for all). There were also no differences in all dimensions of fatigue between groups at each recorded time point (*p* > 0.05 for all). Emotional distress was also similar (*p* = 0.235).

## 4. Discussion

The safety and feasibility of IPMT to better address pain, mobility, and QOL has heretofore never been prospectively addressed in patients with USM, who are at high risk of clinical exacerbation from such interventions. From this exploratory randomized study, the first of its kind to date, IPMT is potentially feasible for this high-risk population, with a clear majority of patients being able to complete the assigned regimen. During the observation period, in the INT group no serious side effects occurred requiring surgical intervention. However, the conclusion about the safety of IPMT can only be made with restrictions, given the high percentage of patients lost to follow-up or death.

It should first be addressed that this study was not powered to evaluate secondary endpoints such as pain response, bone density, pathologic fracture rate (which was imbalanced at baseline), and QOL. Hence, the statistically equivocal results in most of these parameters cannot be used to conclude that IPMT offers no benefit as compared to passive muscle relaxation. Rather, this study demonstrates its safety and feasibility, in efforts to further utilize this regimen in larger studies to adequately test other such endpoints.

This being said, pain response may be less impacted by IPMT and more by RT technique, as shown in promising randomized trials of ablative versus fractionated RT [31,32]. Herein, merging both 3DCRT and IMRT cases would not be expected to confound results, as both are fractionated and do not display differences in relevant endpoints. Bone density changes generally do not occur acutely and may also be impacted by other factors such as the short duration of follow-up herein.

The covariate analysis of pain response during t0–t2 showed no influence of OMED on VAS values in the INT group (*p* = 0.120). However, within the time period t0–t2, pain response within that group was clearly evident (*p* = 0.009). Similarly, examination of the supervised training units from t_0_ to t_1_ showed significant pain relief (*p* < 0.001, data not shown).

Despite the numerically increased initial fracture rate in the INT arm (n = 17, 63% vs. CON: n = 11, 39%, *p* = 0.079) (which also had statistically higher SINS scores at baseline), no *de novo* fractures occurred. Only existing fractures showed a visible increase in INT (1/14 cases) and CON (3/18) (*p* = 0.419).

Future studies should stratify groups according to fracture rate at baseline or SINS score to avoid imbalance between groups. Importantly, the numerically higher pathologic fracture rate in INT at baseline did not translate to appreciable QOL changes, which is noteworthy. Lastly, it is also relatively intuitive that there were largely no significant QOL differences between arms, as QOL is a complex outcome that depends on a multitude of factors such as systemic disease status, ongoing therapies, and baseline functionality. It is thus likely that effects on QOL by IPMT (if present) would be blurred between cohorts based on other factors known to contribute to QOL.

The utility of this investigation impacts future studies in patients with spinal metastases. Historically, patients with stable spine metastases were often restricted from such activities, with even tighter restrictions in USM cases. In the more recent era, many protocols do not specify whether these exercise regimens are allowed. For instance, the RTOG 0631 protocol does not make a specific recommendation on this matter. Although that study pertains to SBRT instead of a traditional 5–10 fraction regimen as utilized herein, further work must be done to verify whether IPMT is safe for well-selected USM cases undergoing SBRT (recognizing that many will not be able to receive SBRT for several reasons).

The difficulty of planning a fixed number of training sessions in the palliative setting is well acknowledged, and as a result there was no precedent to how many high-risk USM patients could complete these sessions. This was a major reason why this randomized trial was exploratory in nature and why formal power calculations could not be made. All patients herein experienced systemic progression at some point during follow-up, which (along with side effects of therapy in itself) often requires temporary or prolonged stationary accommodation and may not be conducive to continuing the training program.

In addition to the above, there are several limitations meriting elaboration. Along with the small sample sizes, short follow-up/patient survival, and single-center nature, studies of the palliative population encompass inherently heterogeneous patients, and the effect on subgroups thus cannot be reliably analyzed. This also makes the results difficult to extrapolate to other work, along with the fact that the particular assessment methods (e.g., VAS) and frequencies thereof may differ from other work, hence also limiting generalizability. Second, corticosteroid doses were not accounted for, which may impact pain levels and “pain flares”. Third, reasons for opioid usage as well as subjective pain relief are inherently difficult to assess, and are known limitations of any palliative study despite the prospective nature. Fourth, because the patients were included with a Karnofsky index >70%, it may not necessarily include a representative population reflective of clinical practice. Nevertheless, these shortcomings do not diminish the requirement to construct similar randomized trials powered for other endpoints, especially given that the safety and feasibility of IPMT in the high-risk USM population has been supported by these randomized results.

## 5. Conclusions

IPMT is potentially feasible for high-risk USM patients. Future trials adequately powered for relevant endpoints are thus recommended.

## Figures and Tables

**Figure 1 cancers-11-01771-f001:**
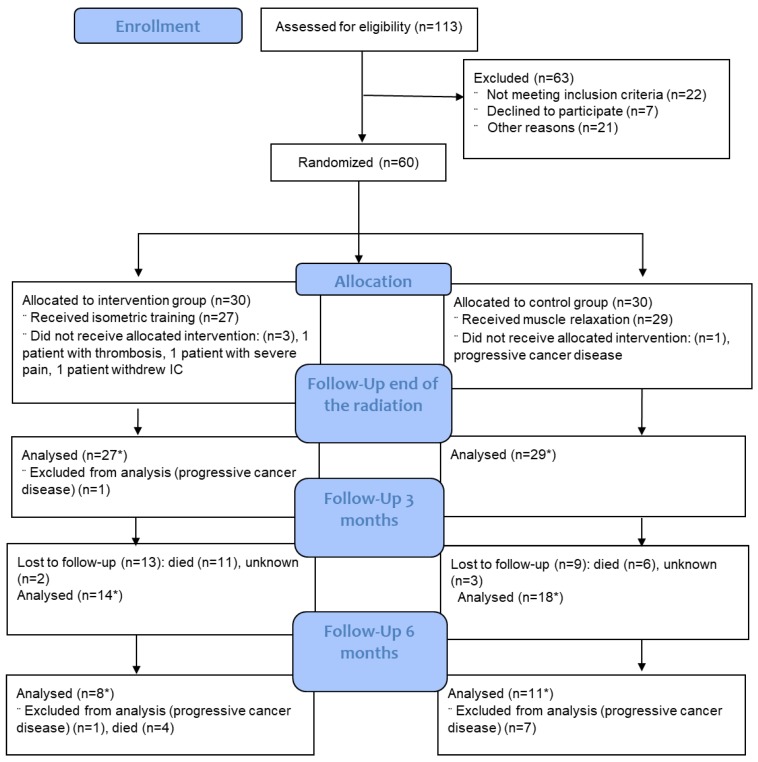
Patient flow through the study. Explanation: * number can differ depending on endpoint, because in some cases individual questionnaires were not collected.

**Figure 2 cancers-11-01771-f002:**
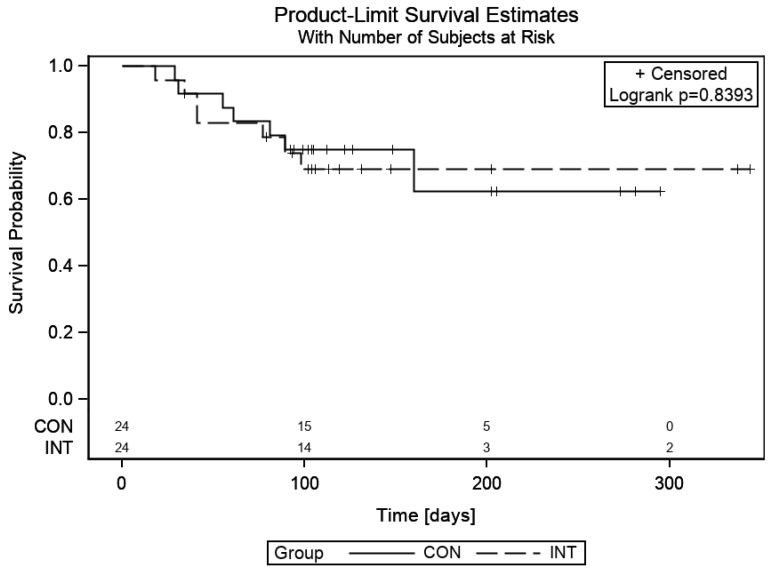
Overall survival of both arms, time in days.

**Figure 3 cancers-11-01771-f003:**
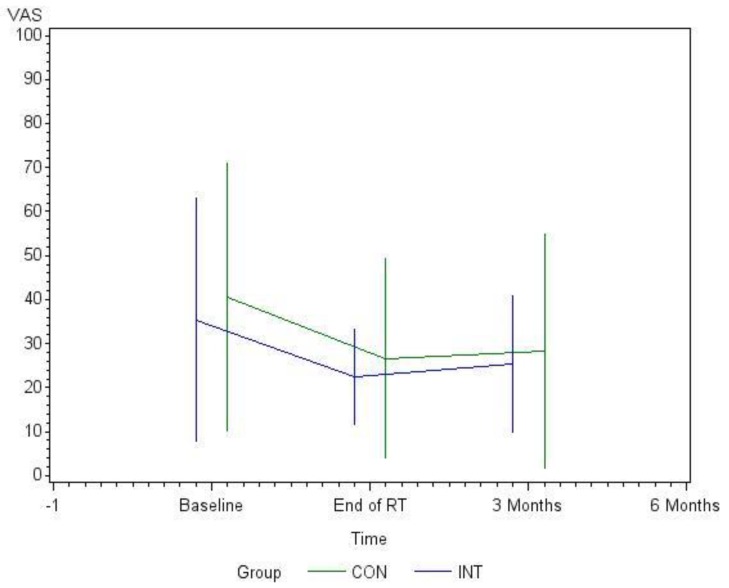
Graphical representation of pain levels using the Visual Analogue Scale (VAS) during the observation period in both groups.

**Table 1 cancers-11-01771-t001:** Demographics. Baseline characteristics of randomly assigned participants.

	Intervention Group n = 27		Control Group n = 29		*p*-Value
	n	%	n	%	
Age (years, mean SD)	62.1 (8.8)		61.1 (8.5)		0.724
*Gender*					0.611
Male	13	48.1	12	41.4	
Female	14	51.9	17	58.6	
Weight (kg, mean SD)	71.5 (11.3)		75.1 (15.4)		0.337
Height (cm, mean SD)	172.2 (8.6)		170.4 (9.0)		0.426
Body mass index (BMI, mean SD, kg/m^2^)	24.4 (4.1)		25.8 (4.6)		0.260
*Primary site*					0.654
Lung cancer	8	29.6	14	48.3	
Breast cancer	7	25.9	6	20.7	
Prostate cancer	4	14.8	2	6.9	
Other	8	29.7	3	24.1	
*Localization metastases*					0.423
Thoracic	20	74.1	24	82.8	
Lumbar	7	25.9	5	17.2	
*Distant metastases at baseline*					
Viszeral	9	33.3	13	44.8	0.379
Lung	5	18.5	11	37.9.1	0.108
Brain	7	25.9	5	17.2	0.423
Soft tissue	2	7.4	7	24.1	0.089
Hormontherapy	9	33.3	6	20.7	0.286
Immunotherapy	12	44.4	13	44.8	0.977
Chemotherapy	19	70.4	21	72.4	0.866
Surgery	15	55.6	12	41.4	0.289
Bisphosphonate at baseline	11	40.7	14	48.3	0.571
Orthopedic corset at baseline	10	37.0	14	48.3	0.396
Mizumoto-Score (mean, SD)	5.0 (2.0)		5.5 (1.7)		0.260
SINS-Score (mean, SD)	12.0 (2.5)		10.3 (2.2)		0.007
*Medication at baseline*					
Sleeping medication	4	14.8	9	31.0	0.151
Psychiatric medication	8	29.6	8	27.6	0.866
Dexamethasone	5	18.5	2	6.9	0.189
Opiate	15	55.6	16	55.2	0.977
NSAID	20	74.1	22	75.9	0.877
Inpatient stay	13	48.1	11	37.9	0.440

Abbreviation: *SD*: standard deviation; *SINS*: spinal instability neoplastic score; *others*: adenoid cystic carcinoma, carcinoma of unknown primary (CUP), gastrointestinal stroma tumor (GIST), melanoma, mesothelioma, pancreatic cancer, renal cancer, thyroid cancer, urothelial carcinoma.

**Table 2 cancers-11-01771-t002:** Pain levels using the Visual Analogue Scale (VAS) at respective points in time of survey in both groups.

		Intervention Group n = 27			Control Group n = 29		*p*-Value
VAS	n	mean	SD	n	mean	SD	
Baseline	27	41.3	29.6	29	44.3	29.3	0.665
End of radiotherapy	26	30.6	19.7	29	29.1	24.8	0.659
3 months	14	25.4	15.5	18	28.3	26.6	1.000
6 months	7	24.3	18.1	9	25.0	26.1	0.830

Abbreviation: *VAS*: visual analog scale.

**Table 3 cancers-11-01771-t003:** These results demonstrate the bone density (HU = Hounsfield units) in metastatic bone before RT (baseline), three and six months after RT. The results were presented by absolute and relative values (%) of HU within and between group as median (Hodges–Lehmann estimate) and IQR.

	Intervention Group			Within Group	Control Group			Within Group	Differences between Groups		
	n	Median	IQR	*p*-Value	n	Median	IQR	*p*-Value	HL	95% CI	*p*-Value
*All metastases*											
HU Baseline	25	200.0	136.0–240.0		29	168.0	139.0–268.0		−3.0	−57.0–47.0	0.903
HU T2	13	278.0	215.0–380.0		18	219.5	137.0–385.0		47.5	−63.0–150.0	0.378
HU T3	8	364.0	237.0–364.0		10	348.5	217.0–478.0		9.0	−144.0–160.0	0.756
*3 months*											
HU T0–T2	13	1.00	−18.0–190.0	0.350	18	30.5	−9.0–70.0	0.101	5.5	−48.0–154.0	0.826
HU T0–T2 (%)	13	0.38	−7.5–118.0	0.491	18	19.0	−7.3–31.7	0.060	3.2	−26.0–82.1	0.674
*6 months*											
HU T0–T3	8	111.5	−23.5–268.5	0.219	10	99.5	69.0–175.0	0.006	−2.0	−137.0–190.0	0.965
HU T0–T3 (%)	8	67.8	−9.0–139.5	0.219	10	49.8	19.4–118.2	0.004	−0.2	−68.7–14.9	0.964
*Subgroup analysis*											
*Osteolyltic metastases*											
HU Baseline	10	128.5	108.0–192.0		15	153.0	124.0–173.0		−17.5	−59.0–32.0	0.506
HU T2	4	268.0	160.5–365.5		6	196.0	112.0–238.0		56.5	−210.0–268.0	0.749
HU T3	3	380.0	136.0–384.0		5	237.0	217.0–323.0		57.0	−238.0–178.0	0.551
*3 months*											
HU T0–T2	4	95.0	−7.5–224.5	0.500	6	32.0	−31.0–43.0	0.469	81.5	−70.0–290.0	0.456
HU T0–T2 (%)	4	59.0	−3.7–166.0	0.500	6	25.4	−7.3–33.3	0.312	53.6	−41.7–221.3	0.594
*6 months*											
HU T0–T3	3	223.0	0.0–259.0	0.500	5	78.0	69.0–122.0	0.125	101.0	−175.0–311.0	0.371
HU T0–T3 (%)	3	138.5	0.0–214.0	0.500	5	56.1	41.1–118.2	0.125	68.8	−145.2–226.3	0.551
*Mixed metastases*											
HU Baseline	15	230.0	186.0–261.0		14	252.5	165.0–349.0		25.5	−52.0–116.0	0.513
HU T2	9	278.0	222.0–448.0		12	317.0	148.5–432.5		44.5	−127.0–162.0	0.696
HU T3	5	348.0	247.0–483.0		5	478.0	376.0–524.0		−41.0	−277.0–158.0	0.834
*3 months*											
HU T0–T2	9	1.00	−18.0–187.0	0.476	12	16.0	−7.0–107.5	0.266	−9.0	−79.0–163.0	0.804
HU T0–T2 (%)	9	0.38	−7.7–71.6	0.652	12	7.9	−5.6–31.4	0.204	0.6	−31.3–72.0	0.972
*6 months*											
HU T0–T3	5	−15.0	−32.0–278.0	0.812	5	114.0	85.0–260.0	0.062	−82.0	−310.0–227.0	0.531
HU T0–T3 (%)	5	−5.7	−12.4–135.6	0.812	5	43.5	19.4–115.6	0.062	−31.7	−128.2–116.2	0.676

Abbreviations: HU = Hounsfield units, IQR = interquartile range, T_0_ = baseline, T_2_ = 3 months, T_3_ = 6 months, T_0_–T_2_ = difference baseline minus 3 months, T_0_–T_3_ = difference baseline minus 6 months. HL = Hodges-Lehmann estimator, 95% CI = 95% Confidence Interval.

**Table 4 cancers-11-01771-t004:** Effects on QOL (EORTC QLQ-BM 22) questionnaire.

Symptom Scales						
	Intervention Group			Control Group		
**Painful sites**	n	mean	SD	n	mean	SD
End of radiotherapy (t1)	26	29.5	19.5	29	20.7	20.3
3 months (t2)	14	27.6	19.9	18	22.2	13.9
6 months (t3)	7	32.4	18.4	9	17.8	14.1
Treatment effect (t0–t2) after 3 months *p* = 0.478, (t0–t3) after 6 months *p* = 0.753						
Effect size (t0–t2) after 3 months −5.9, (t0–t3) after 6 months 4.7						
**Pain characteristics**						
End of radiotherapy (t1)	26	44.9	28.4	29	36.0	32.6
3 months (t2)	14	30.2	28.7	18	29.5	28.3
6 months (t3)	7	22.2	22.2	9	23.5	26.3
Treatment effect (t0–t2) after 3 months *p* = 0.813, (t0–t3) after 6 months *p* = 0.470						
Effect size (t0–t2) after 3 months −2.6, (t0–t3) after 6 months −10.8						
**Functional interference**						
End of radiotherapy (t1)	26	44.6	24.6	29	36.2	22.6
3 months (t2)	14	37.8	29.3	18	28.5	18.7
6 months (t3)	7	28.6	26.8	9	31.9	19.8
Treatment effect (t0–t2) after 3 months *p* = 0.611, (t0–t3) after 6 months *p* = 0.269						
Effect size (t0–t2) after 3 months −3.8, (t0–t3) after 6 months −9.9						
**Psychosocial aspects**						
End of radiotherapy (t1)	26	58.5	17.5	29	50.4	18.1
3 months (t2)	14	52.4	20.8	18	52.2	18.6
6 months (t3)	7	42.9	23.3	9	51.2	21.5
Treatment effect (t0–t2) after 3 months *p* = 0.072, (t0–t3) after 6 months *p* = 0.348						
Effect size (t0–t2) after 3 months −15.2, (t0–t3) after 6 months −7.2						

**Table 5 cancers-11-01771-t005:** Effects on fatigue according to EORTC QLQ-FA 13 questionnaire.

	Intervention Group			Control Group		
**Physical fatigue**	n	mean	SD	n	mean	SD
End of radiotherapy (t1)	25	60.0	25.1	29	52.0	28.9
3 months (t2)	14	45.2	31.0	18	50.0	28.7
6 months (t3)	7	45.2	34.6	9	46.3	30.1
Treatment effect (t0–t2) after 3 months *p* = 0.06, (t0–t3) after 6 months *p* = 0.06						
Effect size (t0–t2) after 3 months −16.2, (t0–t3) after 6 months −16.2						
**Emotional fatigue**						
End of radiotherapy (t1)	25	39.7	30.0	29	29.9	28.3
3 months (t2)	14	27.4	23.7	18	31.5	30.2
6 months (t3)	7	31.0	39.9	9	30.6	30.3
Treatment effect (t0–t2) after 3 months *p* = 0.056, (t0–t3) after 6 months *p* = 0.928						
Effect size (t0–t2) after 3 months−18.6, (t0–t3) after 6 months −1.6						
**Cognitive fatigue**						
End of radiotherapy (t1)	25	17.3	18.7	29	13.0	16.8
3 months (t2)	14	16.7	16.2	18	13.0	18.4
6 months (t3)	7	23.8	37.1	9	13.6	12.1
Treatment effect (t0–t2) after 3 months *p* = 0.117, (t0–t3) after 6 months *p* = 0.440						
Effect size (t0–t2) after 3 months −10.8, (t0–t3) after 6 months 6.9						
**Interference with daily life**						
End of radiotherapy (t1)	25	50.7	37.4	29	41.4	29.1
3 months (t2)	14	50.0	36.4	18	28.9	30.8
6 months (t3)	7	42.9	41.8	9	29.6	35.1
Treatment effect (t0–t2) after 3 months *p* = 0.780, (t0–t3) after 6 months *p* = 0.230						
Effect size (t0–t2) after 3 months −2.9, (t0–t3) after 6 months 16.9						
**Social sequelae**						
End of radiotherapy (t1)	25	14.7	23.7	29		
3 months (t2)	14	16.7	28.5	18		
6 months (t3)	7			9		
Treatment effect (t0–t2) after 3 months *p* = 0.936, (t0–t3) after 6 months *p* = 0.366						
Effect size (t0–t2) after 3 months 0.79, (t0–t3) after 6 months 12.2						

**Table 6 cancers-11-01771-t006:** Effects on emotional distress according to FBK-R10 questionnaire.

	Intervention Group			Control Group		
**FBK R10**	n	mean	SD	n	mean	SD
Baseline (t0)	26	23.7	8.5	29	17.9	9.2
End of radiotherapy (t1)	26	19.7	8.5	28	15.7	8.6
3 months (t2)	14	19.7	9.3	18	13.8	9.4
Treatment effect (t0–t2) after 3 months between the groups *p* = 0.235,			
Effect size (t0–t2) after 3 months −4.1

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
