# Peer review of "Paravertebral Muscle Training in Patients with Unstable Spinal Metastases Receiving Palliative Radiotherapy: An Exploratory Randomized Feasibility Trial"

_cancers, 2019, doi:10.3390/cancers11111771_

Round 1
Reviewer 1 Report
The authors present a safety and feasibility study of IPMT in patients with unstable spinal metastases using a randomized controlled trial approach. The authors should be applauded for conducting this novel and resource intensive study. There are however a couple of issues that should be clarified by the authors to be able to interpret the results in the correct way and translate them into clinical practice. Furthermore, the conclusion of the authors is strong despite the numerous limitations of the study.
Introduction
the authors are right to mention that patients with unstable metastases may not always be good surgical candidates. However, they state that palliative radiotherapy is also an effective treatment in patients with unstable metastases. They refer to many studies regarding all bone metastases and also mainly studies without information regarding stability. As such the authors should amend this sentence. It is unclear if the authors think that IPMT could be an option in addition to RT for patients with unstable spinal metastases for whom surgery is not an option for several reasons or if they think IPMT should be used for every patients with unstable metastases and may even replace surgery. Please commentMaterials and Methods
Information regarding the inclusion and exclusion criteria is limited. The authors should expand on the eligibility of patients. Were patients referred for surgical evaluation? Were they not surgical candidates? Were patients offered other treatment options? Only patients with a karnofky score of 70 or higher were included, although for a pilot study this makes sense the authors should recognize that unstable metastases often results in significant functional impairment and therefore excluding a lot of patients. They mention spinal cord compression as an exclusion criteria, how was this defined? Bilsky? It is unclear if every patient received follow-up imaging as part of the protocol or if this based on a clinical decision.Results
The authors fail to report important treatment information. They do not report information regarding radiotherapy details per group. Please provide this info. The loss to follow-up is really high, although this is partly inherent to the patient population this is highly influences the results. Also, in general a three month life expectancy is considered necessary to be able to undergo surgery which in this case may indicate that these patients were not surgical candidates at the beginning. In itself this is not a problem however the authors should clearly describe what kind of patients we are talking about. The legend in figure 2 says that time is in months however this seems incorrect considering a follow-up time of 300 months would correspond to a follow-up of multiple years. The number of patients completing the exercises is low although the authors make it look it is high by presenting as the percentage of patients that is alive. The baseline pain scores of the patients is relatively low for patients with unstable spinal metastases as compared to other studies this raises questions as to what kind of patients we are talking about. The authors state that there is no difference in OMED at the end of RT and during follow-up, however were there any changes made at the beginning of radiotherapy? Any addition of steroids? etc? The authors describe a temporal relationship for the pain response which is nothing new, although the authors present this info in a graph I think this information is better suited in a table. It is not clear from the graph what the actual numbers are. Overall the numbers in the groups are really low over time which makes sense, however it questions the validity of the analyses performed and the presentation of the P-values.Discussion
The authors conclude that IPMT is safe and feasible for this patient population. Yet, this conclusion seems a bit too strong for the results they show. There was a high loss to follow-up which is not recognized in the feasibility and safety rate. Furthermore, there is limited information regarding the spinal instability of these patients which highly limits the generalizability of the results. The authors state in the discussion that the supervised training units showed significant pain relief. This is important data however this is not shown.General comments
Although not addressed and maybe also not known it would be really interesting to know what the patient experience is with these exercises. To emphasize the authors should be really careful with their conclusion, is seems that they say that IPMT is a safe option for all patients with unstable spinal metastases however based on their study this conclusion cannot be drawn and this must be recognized.
Author Response
We greatly thank the reviewers for their time in reviewing the manuscript and for the positive comments. We have worked to address each facet of the revision suggested by the reviewer and are happy to address any further issues at any point in time. Thanks very much.
Reviewer #1
The authors present a safety and feasibility study of IPMT in patients with unstable spinal metastases using a randomized controlled trial approach. The authors should be applauded for conducting this novel and resource intensive study. There are however a couple of issues that should be clarified by the authors to be able to interpret the results in the correct way and translate them into clinical practice. Furthermore, the conclusion of the authors is strong despite the numerous limitations of the study.
Thank you for the kind words. We have reviewed the entire manuscript again for clarity and readability.
Introduction
the authors are right to mention that patients with unstable metastases may not always be good surgical candidates. However, they state that palliative radiotherapy is also an effective treatment in patients with unstable metastases. They refer to many studies regarding all bone metastases and also mainly studies without information regarding stability. As such the authors should amend this sentence.
Thank you for this comment. We revised this sentence (line 58-59), although we consider the work cited to be relevant in this context since it is nearly certain that those studies contained a large proportion of both stable and unstable metastases.
It is unclear if the authors think that IPMT could be an option in addition to RT for patients with unstable spinal metastases for whom surgery is not an option for several reasons or if they think IPMT should be used for every patients with unstable metastases and may even replace surgery. Please comment
Thank you for critical comment. Isometric training was not intended as a replacement for surgical or conservative intervention, and rather provides an additional measure to improve QOL. We have made the necessary clarifications (line 63).
Materials and Methods
Information regarding the inclusion and exclusion criteria is limited. The authors should expand on the eligibility of patients. Were patients referred for surgical evaluation? Were they not surgical candidates? Were patients offered other treatment options?
Prior to RT, an interdisciplinary surgical examination was performed to determine operability and necessity thereof. In some cases there was inoperability from extensive comorbidities and no fixation options with extensive osseous metastasis. In other cases, the patients refused surgical stabilization. We added this information in lines 89-90.
Only patients with a karnofky score of 70 or higher were included, although for a pilot study this makes sense the authors should recognize that unstable metastases often results in significant functional impairment and therefore excluding a lot of patients.
Thank you for this comment; we have made the corresponding additions (line 319-321).
They mention spinal cord compression as an exclusion criteria, how was this defined? Bilsky?
Thank you for this comment, We have modified the text accordingly (line 96).
It is unclear if every patient received follow-up imaging as part of the protocol or if this based on a clinical decision.
Thank you for this comment. The study design was intended to provide an imaging follow-up using CT at 3 and 6 months following RT.
Results
The authors fail to report important treatment information. They do not report information regarding radiotherapy details per group. Please provide this info.
We added information about RT in lines 113-121.
The loss to follow-up is really high, although this is partly inherent to the patient population this is highly influences the results. Also, in general a three month life expectancy is considered necessary to be able to undergo surgery which in this case may indicate that these patients were not surgical candidates at the beginning. In itself this is not a problem however the authors should clearly describe what kind of patients we are talking about.
Thank you for this comment. We have made the corresponding additions per the above comment (line 89-90).
The legend in figure 2 says that time is in months however this seems incorrect considering a follow-up time of 300 months would correspond to a follow-up of multiple years.
Thank you for this comment. We apologize for the inaccuracy and corrected Figure 2.
The number of patients completing the exercises is low although the authors make it look it is high by presenting as the percentage of patients that is alive.
The baseline pain scores of the patients is relatively low for patients with unstable spinal metastases as compared to other studies this raises questions as to what kind of patients we are talking about.
We have made the appropriate additions (as noted above, line 89-90).
The authors state that there is no difference in OMED at the end of RT and during follow-up, however were there any changes made at the beginning of radiotherapy? Any addition of steroids? etc?
The relevant information (corticosteroids) is given in Table 1.
The authors describe a temporal relationship for the pain response which is nothing new, although the authors present this info in a graph I think this information is better suited in a table. It is not clear from the graph what the actual numbers are.
The pain levels are shown in Table 2; this graph was designed to illustrate the simultaneous course of pain.
Overall the numbers in the groups are really low over time which makes sense, however it questions the validity of the analyses performed and the presentation of the P-values.
We agree with you on this point, although it should be noted that the p-values for these low number of patients were intended to be presented in a descriptive manner.
Discussion
The authors conclude that IPMT is safe and feasible for this patient population. Yet, this conclusion seems a bit too strong for the results they show. There was a high loss to follow-up which is not recognized in the feasibility and safety rate. Furthermore, there is limited information regarding the spinal instability of these patients which highly limits the generalizability of the results.
Thank you for your comments, we have revised the conclusion.
The authors state in the discussion that the supervised training units showed significant pain relief. This is important data however this is not shown.
Thank you for your comment. The VAS levels were collected directly in the lead radiation oncology department and are shown in Table 2. However, the patients made VAS assessments during the supervised training in the cooperating sports medical unit. In the final evaluation we found similar but not identical VAS levels. Therefore, in order to not confuse the reader, we did not present this information.
General comments
Although not addressed and maybe also not known it would be really interesting to know what the patient experience is with these exercises.
We have consistently received positive feedback from patients. However, this was not reflected in the QOL questionnaires.
To emphasize the authors should be really careful with their conclusion, is seems that they say that IPMT is a safe option for all patients with unstable spinal metastases however based on their study this conclusion cannot be drawn and this must be recognized.
Thank you for your comments, we have revised the conclusion.
Reviewer 2 Report
very well written paper with a well described study. My only comment is to provide references for the end of the first paragraph in the Background section
Author Response
We greatly thank the reviewers for their time in reviewing the manuscript and for the positive comments. We have worked to address each facet of the revision suggested by the reviewer and are happy to address any further issues at any point in time. Thanks very much.
very well written paper with a well described study. My only comment is to provide references for the end of the first paragraph in the Background section.
Thank you for the kind words. We have completed the references.
We have corrected an error in section 3.2. We apologize for the inaccuracy.
“In INT, 18 (67%) patients completed ≥80% of the planned training sessions. The mean total number of completed training units was 7.8 (SD 14.8 3.3), and the mean number of potentially feasible units was 10.1 (SD 2.1).”